# How Good Is our Place—Implementation of the Place Standard Tool in North Macedonia

**DOI:** 10.3390/ijerph17010194

**Published:** 2019-12-27

**Authors:** Dragan Gjorgjev, Mirjana Dimovska, George Morris, John Howie, Mirjana Borota Popovska, Marija Topuzovska Latkovikj

**Affiliations:** 1Institute of Public Health of the Republic of North Macedonia, 1000 Skopje, North Macedonia; m.dimovska2016@gmail.com; 2European Centre for Environment and Human Health, University of Exeter Medical School, Truro TR1 3HD, UK; geomorris55@hotmail.co.uk; 3NHS Health Scotland, Meridian Court, 5 Cadogan Street, Glasgow, Scotland G2 6QE, UK; john.howie@nhs.net; 4Institute for Sociological, Political and Juridical Research, Skopje, University “Ss. Cyril and Methodius” 1000 Skopje, North Macedonia; mborota@isppi.ukim.edu.mk (M.B.P.); marija_t@isppi.ukim.edu.mk (M.T.L.)

**Keywords:** place, place standard, community engagement, community empowerment, health determinants, wellbeing

## Abstract

This study describes the implementation, in North Macedonia, of a “tool”, initially devised in Scotland, to generate community and stakeholder discussion about the places in which they live and notably a place’s capacity to generate health wellbeing and greater equity among citizens. In this study, the “place standard tool” (PST) is viewed from the perspective of creating places which can deliver a triple win of health and wellbeing, equity, and environmental sustainability. Skopje, North Macedonia’s capital, inevitably differs economically, culturally, and politically from Scotland, thus providing an opportunity to augment existing knowledge on adaptability of the tool in shaping agendas for policy and action. Тhe PST was tested through seminars with selected focus groups and an online questionnaire. Over 350 respondents were included. Information on priorities enabled the distillation of suggestions for improvement and was shared with the Mayor and municipal administration. Skopje citizens valued an approach which solicited their views in a meaningful way. Specific concerns were expressed relating to heavy traffic and related air and noise pollution, and care and maintenance of places and care services. Responses varied by geographic location. Application of the PST increased knowledge and confidence levels among citizens and enthusiasm for active involvement in decision making. Effective implementation relies heavily on: good governance and top-level support; excellent organization and good timing; careful training of interviewers and focus group moderators; and on prior knowledge of the participants/respondents.

## 1. Introduction

Urban environments are home to two thirds of the population of the European region, providing opportunities for individuals and families “to prosper and promote health through enhanced access to services, culture and recreation” [1]. Yet, alongside economic prosperity, cities can concentrate poverty and ill health, social isolation, and high levels of pollution. Within cities, environmental exposure is often socially patterned. When compared to more affluent neighborhoods, socially disadvantaged areas are often characterized by a concentration of environmental risks and a lack of amenities. These contrasting circumstances denote an absence of environmental justice, whose component parts are ‘geographical’ (or ‘distributional’) justice and so-called ‘procedural’ justice. Procedural justice concerns the lack of access to, and participation in, decision making processes and procedures. Such deficits often characterize socio-demographically disadvantaged groups [2,3,4]. Another striking feature of many urban environments is the “take-make-consume-dispose” model which underpins economic growth. This emphasizes the pursuit of convenience and promotes a disconnectedness from nature which, in turn, drives unhealthy and unsustainable lifestyles. The transition to more sustainable societal and individual behaviors demands integrated approaches at a central and local level, to which governments, citizens, and the private sector are all key contributors [5].

Natural and built environments of good quality have positive effects on our physical and mental health and on social wellbeing [5]. Extensive scientific evidence supports the theory that urban health is directly affected by a combination of physical and social factors, and the availability of, and access to, critical services such as health and social care. The built environment (connectivity of street networks, aesthetic qualities of place, mixed land use etc.) may act to reduce physical activity that will lead, among other things, to increased rates of cardiovascular diseases and increased all-cause mortality [6]. Not only physical health, but also mental health concerns (e.g., psychological distress and impaired child development) are affected by a poor built environment. Urban design can impact on the ability or inability to choose and access healthy lifestyle opportunities or engage positively in healthy behaviors. Urban design is also linked to levels of crime and violence [7,8,9,10,11,12].

The place standard tool (PST) has its origins in Scotland, where a Scottish policy position on place and health has evolved largely in response to two documents, Good Places Better Health (2008) and Creating Places (2013). Both publications set out the pivotal role that effective collaboration between communities, public, voluntary, and private sectors has in generating the range of high quality places critical to ensuring better life outcomes for all [13,14]. The PST itself was developed through collaboration between NHS health Scotland, Scottish government, architecture and design Scotland, and Glasgow city council [15]. The tool is a simple framework to structure conversations about place. It allows those with an interest in and influence upon a particular place to think about the physical (for example its buildings, spaces, and transport links) as well as the social aspects (for example whether people feel they have a say in decision making).

By providing prompts for discussion, the tool allows all the elements of a place to be considered in a methodical way. Importantly, it pinpoints the assets of a place, as well as areas where a place could improve.

As a consortium member working on the EU-funded INHERIT project (intersectoral health and environment research and innovation project), the institute of public health of RN Macedonia set an objective to implement the place standard tool (PST). The initiative, applied in the city of Skopje, was one of 15 case studies selected for evaluation in terms of their capacity to deliver the triple win of improved health and wellbeing, equity, and environmental sustainability. The 15 case studies were chosen from over 100 European examples held on an INHERIT database. In addition to its contribution within the overall objectives of the INHERIT project, a key aim of the Skopje case study was to understand the applicability and relevance of the PST in a context which inevitably differs economically, socially, culturally, and politically from Scotland, where the tool originates [13]. In doing so, the study seeks to determine not only the potential for wider application across North Macedonia, but to add to a growing body of international evidence regarding the flexibility and utility of the PST across regions. A secondary aim was to develop a sense of how the PST can be used in RN Macedonia to improve the processes of intersectoral cooperation, in this context, the cooperation among residents and the many players, who are important in understanding and shaping the places where people live. This might include different branches within municipal government.

The four-year INHERIT project is a research project that aims to understand how lifestyles and behaviors can be changed in order to achieve a triple win of promoting health, environmental sustainability, and equity in the areas of living, moving, and consuming. The INHERIT project goals reflect the strategic goals of the WHO healthy cities network. Here, international cooperation and working links between European cities and movement is encouraged, as is the promotion of partnerships among agencies concerned with urban issues. Through such an approach, it is considered that expert knowledge, effective methods, high quality evidence, and practice expertise can be brought together to promote and advocate for health [16]. The institute of public health considered the PST tool to be potentially a powerful instrument for achieving those objectives.

Today, the PST is applied across all municipalities in Scotland, is supported by a national governance framework, and a network of place standard leads in each municipality and in both of Scotland’s national parks. Its application internationally has extended across 11 countries in Europe, and in 2020 a second version of the place standard will be published in due course. Further versions relating to places for children and for the use by those engaged in designing places are also anticipated [17].

This paper aims to present how the PST can help to identify the needs and priorities of the citizens, and how, set against the different contexts, it can contribute and lead to better inter-sectoral cooperation when implementing urban policy. It actually describes the application of a relatively new tool for securing the involvement of communities in a comprehensive discussion of the merits and demerits of the places they live, and particularly the influence this has on matters of health, wellbeing, and equity, and through this, plotting the direction of change.

## 2. Materials and Methods

The PST tool has been piloted at city level as part of INHERIT, in the Republic of North Macedonia (RN Macedonia) and in Latvia in 2018–2019. This involved the cities of Skopje and Riga. This paper describes the approach and presents results from the city of Skopje study. The city of Skopje, RN Macedonia, as the RN Macedonia capital, is not only a major administrative and political center, but also the key educational center in the country. Almost a third of the total population live in Skopje metropolitan (627,558 population) [18] and many more people work and commute to the city on a daily basis. This creates a variety of pressures on the environment through increased demand for food, water, road, and public transport networks, and increased energy demand and consumption. With an estimated GDP per capita higher than the national average of 6083 USD (in 2018), the city of Skopje is still behind the GDP average of the capital cities of EU (EU—28 countries with 36,513.7 USD, including the United Kingdom with a GDP per capita of 42,491.4 USD) [19,20].

The data collection techniques used during the study were: online survey, interviews, and focus groups on a purposive research (sub) samples.

### 2.1. Online PST Questionnaire/Survey

As part of a two-stage approach, the PST online questionnaire was posted for two months on the Karpos municipality official website (July–August 2018) and on the city of Skopje’s website from July to September 2019. The survey was anonymous and people who live/work in the territory of the municipality of Karpos and the city of Skopje, respectively, responded to the electronic survey on their own initiative (n = 278 respondents). To maximize reach, the project was actively promoted by the local and city authorities on their official websites.

### 2.2. Focus Group Work

Two focus group sessions were conducted using pre-selected inclusion criteria that reflected the demographic characteristics of the municipality of Karpos. To reflect the heterogeneity of the target population, inclusion criteria included gender, age, ethnicity, rural/urban residence, and educational and socio-economic status. Sixteen respondents selected (out of 72 candidates) completed the public online-application form to participate in the focus groups, which was posted on the website of the institute for sociological, political and juridical research at the State University. The other focus group involved representatives from the municipal administration and included the mayor. This group discussed the level of efficiency and quality of collaboration among sectors during the PST implementation. Each focus group session began with an explanation of the project goals, key research objectives, and the tool, and was followed by a moderated group discussion on each topic of the tool. Each participant signed a consent form and a sign-in sheet.

### 2.3. The Instrument

The place standard tool can be described as “a flexible product that translates complex public health and place-making theory into a simple tool that supports communities, organizations, and businesses to work together and identify both the assets of a place and areas deemed priority for improvement within places that are well-established, undergoing change, or still being planned” [21].

Fourteen readily-understandable questions were developed, covering physical and social aspects and elements of the living space. These are supported by additional sub-questions to help respondents to evaluate each topic, and to assess the place while thinking about the needs of population groups differentiated by age and other characteristics (disability, sexual orientation, parents, etc.) (Table 1). The tool is designed to provide both a quantitative assessment (based on a score of 1 to 7 for each theme) and qualitative response through a free text. The quantitative scores (displayed on a compass diagram) allow, at a glance, an immediate understanding of which dimensions of the place under consideration work well (a score of 7 is the highest possible) and which areas require improving (a score of 1 is the lowest) [15,21].

The translation of the place standard from English to the local language, Macedonian, required minor adaptations to the terminology but this did not result in any substantive changes from the original version published in English. Translation of Table 1 in Macedonian is enclosed in the Appendix A. Ethical approval from the national ethical board was obtained at the initial phase of implementation.

### 2.4. Data Analysis

Both quantitative and qualitative analyses were applied. Quantitative data were analyzed by calculating the mean/arithmetic mean of the responses obtained for each of the questions, on a scale from 1—least satisfied to 7—completely satisfied; and by frequencies, by means of a percentage distribution of answers for each question. At the beginning of the quantitative analysis, a Cronbach’s alpha test (α = 0.892) was performed, indicating high reliability, i.e., internal consistency of the scale for all 14 dimensions.

Qualitative data were analyzed through thematic analysis, for which, at the beginning of the analysis, a protocol of coding the answers into positive and negative was made for each of the 14 aspects of the living place. The remaining open questions for identifying problems and identifying proposed measures/activities were similarly analyzed. For open questions requiring prioritization/ranking, a data quantification protocol was used. More specifically, the used words and phrases from the open questions were grouped into different categories describing the same context or issue, and later ranged according to their frequency and prioritization.

All of the data were statistically analyzed using the IBM statistical package for the social sciences (IBM SPSS Statistic for Windows, Version 19.0. Amonk, NY: IBM Corp.). The summary results from quantitative analysis are illustrated in a compass diagram, created in Excel.

## 3. Results

Table 2 summarizes demographic characteristic of the respondents. Females represent two thirds (62.2%) of total population who responded on the online questionnaire. Regarding the ethnicity of respondents, the research sample for the open-access online survey consisted predominantly of citizens with Macedonian ethnicity (88.8%) and citizens from other ethnicities living in the city (11.2%). In terms of age, most were in the age group of 35 to 44 (38.5%), followed by the age group 25 to 34 (31.7%). The highest respondent rate was achieved in three municipalities—Karpos, Centar and Aerodrom. These are the most developed and attractive urban municipalities of the city of Skopje.

### 3.1. Quantitative Evaluation of Online Survey

Performing the Cronbach’s Alpha test, a high score was calculated (α = 0.892), indicating high reliability (internal consistency of the scale) for all 14 dimensions (Table 3).

Figure 1 presents the summarized scores/arithmetic means obtained from an online questionnaire that varies from *m* = 2.3 to *m* = 4.1. From the online questionnaire, the least rated aspects of living in the city of Skopje are: ‘Traffic and Parking’, followed by ‘Influence and Sense of control’ and ‘Care and maintenance’, while the highest rated aspects are ‘Facilities and amenities’, ‘Housing and communities’ and ‘Identity and belonging’ with the same score, and ‘Feeling safe’.

Detailed information on the frequencies of the scores for fourteen topics of the PST online questionnaire are presented in Table 4. Eight topics are in the negative interval (score < 3.5).

### 3.2. Quantitive Evaluation of Focus Group Work

Rating of place standard topics according to focus group participants is presented in Figure 2. ‘Identity and belonging’ is the highest rated topic (mean score = 5.1), followed by ‘Facilities and amenities’ and ‘Feeling safe (mean score = 4.9)’. The lowest rated were ‘Traffic and parking’ and ‘Care and maintenance’ (mean score = 2.9) (Figure 2).

### 3.3. Comparisons of the Results Obtained through the Online PST Questionnaire vs. Citizens Focus Groups

Differences in rating of the topics were revealed between the analysis of online responses and those of citizens participating in the small and selected focus groups described in methodology. Summarized results from both approaches are shown in Table 5. Only four topics are in the negative interval (score < 3.5) using the focus group approach.

## 4. Discussion

### 4.1. Top Rated Topics from the Online Questionnaire

Access to facilities and amenities (schools, nurseries, libraries, shops, places to meet friends, etc.) plays an important role in our lives and support healthy and fulfilling lives. Healthy, fulfilling places should meet a variety of different needs for learning, health, shopping, and relaxation, and be affordable and accessible for people from different societal groups, whether defined by age, ethnicity, sex, sexuality, and/or disability. While the basic physical characteristics of place are important, so too is the level of maintenance if they are to be used to their full potential. In this pilot-study, the topic facilities and amenities was accorded the highest score (mean = 4.1), while at the same time, the lowest score of 2.8 was given to care and maintenance.

While the qualitative analysis of the answers generated positive comments on the existence of many facilities in the capital, a lack of libraries, reading rooms, and kindergartens was identified. Negative comments were mostly related to concerns over of the physical access to public buildings, highlighting that the mere existence of sufficient facilities does not mean there will equal access for all. Irrespective of whether access/entrance arrangements comply with existing laws and regulations, other issues frequently conspire to deny or prevent access, especially for people with disabilities (notably, wheelchair users) and parents with infants in a stroller. An example of an impediment to access might be blockage of entrances due to parked cars.

An important message from the above is that securing healthy, sustainable, more equitable places through policy and action is about much more than providing facilities and acting to address hazards and risks, it is about understanding the attitudes and beliefs of those who live there, how they feel, and the impediments and barriers they perceive in living a healthy fulfilling life. The INHERIT project confronts this reality and the policy challenges it implies. Specifically, the INHERIT model, which underpins the project, integrates behavior change theory [22] with an established conceptual model from the field of environmental health, which reflects both social and ecological complexity [23]. Indeed, it can be differentiated from other models in environmental health in that it integrates social and ecological complexity as well as behavioral issues, and how this can be addressed through policies and actions. In this way, it is a useful policy tool which integrates the pursuit of the triple win of health and wellbeing environmental sustainability and equity [2]. The INHERIT model explores how multiple interacting driving forces lead to environmental pressures and states, and how these pressures affect different groups of people and their health and well-being. It can also help to determine what measures can be taken to address such behaviors. Since the PST gives the municipality a better understanding of resident’s specific grievances, it can begin to explore how to address these in ways that can also address specific drivers of environmental degradation and ill health.

In our study, housing and community was scored at 3.9, which locates the tendency of the answers in the positive axis interval. The qualitative analysis was aimed at generating answers as to whether there is a choice of quality homes for living and renting, and whether the homes are functioning well. Is housing suitable for all categories of citizens? The city of Skopje and its municipalities are attractive places to live according to most of the respondents. There is a choice of quality residential buildings for different categories of the population who are well organized. The negative aspects include overcrowded places without adequate standards for privacy and a good quality of life. Also highlighted is the lack of an adequate associated infrastructure, characterized by narrow streets, sidewalks and paths, poor or insufficient parking lots, badly drained streets, poor water pressure, etc.

The concept of belonging is not just about membership, rights, and duties, as in the case of citizenship, or simply about forms of identification with groups or other individuals. Anthias (2009) has observed it is also about the ways in which “social place has resonances on stability of the self, on feelings of being part of a larger whole and the emotional and social bonds that are related to such places” [24]. In an attempt to rate the feeling of identity and belonging of the citizens, the PST tool requires respondents to say whether they see the place positively, whether they know and celebrate the history or culture and heritage of the place, and whether they feel connected with the neighbors and community. Here, the meaning of “belonging” relates to the feeling of acceptance as part of a community, feeling safe within it, and having a stake in the future of the “community of membership”. While the mean city score was 3.9, it is necessary to recognize the considerable difference in attitude between citizens living in the older municipalities like Karpos compared to a newer municipality, such as the municipality of Center. Notably, residents in Karpos register a strong sense of identity with the area where many homes are privately owned. In contrast, residents of Center, feel less connection with their area and rented homes predominate. In these areas, the feeling of identity is weaker mostly due to the problem of overcrowding and the continual changes in the resident population. Respondents feel that the new settlers are reluctant to accept longstanding social norms in the municipality and that the neighborhood’s history and cultural heritage are not well understood or respected. Moreover, the new neighborhood configuration can leave the elderly feeling lost. The neighborhood’s history and cultural heritage are not well known to many residents and properly marked. In comparison, the overall score on identity in a Scottish study was 4.0, generated mostly from the responses of citizens who live in the more affluent municipalities of Hillhead and Hardstones, while others municipalities were characterized by negative comments in terms of the issue of drugs misuse and a lack of connectivity between both municipalities [25].

An Australian study found that rural dwellers have higher place dependence compared to the urban dwellers. Again, positive pro-environmental behavioral intentions tend to be recorded in areas where there is high place attachment. Place dependence and place identity are highly correlated with environmentally positive behavior [26,27,28].

### 4.2. Lowest Rated Topics from the Online Questionnaire

Multiple studies have modeled congestion in urban areas and assigned economic values to the excess fuel consumption and time wasted in traffic, concluding that congestion leads to annual economic burdens ranging from $83 billion (Texas, USA) to $124 billion in Canada [29,30,31]. Alternatively, some studies evaluate the public health impact due to exposures to PM_2.5_, or exposure to traffic-related noise and related health outcomes (cardiovascular disease, hypertension, lung cancer, auditory and non-auditory effects, effects on the mental health or impairment of cognitive performance in school children) [32,33,34,35]. Traffic and parking arrangements that allow people to move around safely can help people to get the most out of a place [15]. The city of Skopje’s pilot-study provided an opportunity to assess whether they have a feeling that cars are prioritized instead of people in the places they live; and whether there is any related impact on peoples’ health and wellbeing, and how residents perceived this risk. In addition, respondents were asked whether there is any traffic calming measure implemented to benefit the community.

The topic traffic and parking was the lowest rated topic in the survey (mean = 2.4) that locates the tendency of the answers in the negative axis interval. Poor level of traffic culture, poor implementation of the law and regulations, and constant migration of people towards the capital city, confronting them for the first time with high level of traffic density were identified as the main drivers for this situation. People feel powerless, unsafe, and cars take precedence over against peoples’ needs for more greenery, cleaner air, and peaceful neighborhoods. A newly identified problem, especially on the larger boulevards, is the high speed of driving, especially at night and the noise associated with that. The scores and comments by wheelchair users and parents with infants of small children reflect that they are most dissatisfied. People feel that local authorities “invite” more cars in the cities through actions such as expanding the streets and building parking lots and multi-story car parks in place of parks and greenery. Respondents claimed a complete absence of traffic-calming measures in their living environment. No positive comments were registered on this topic, but there were plenty of suggestions on how to improve the situation.

While the respondents had many suggestions, they did not necessarily feel their voices would be heard. However, an active participation of people in decision-making processes at any level is a crucial component in the process of achieving better places, as is the commitment of the local or central authorities. Discourses on participation are full of spatial metaphors, whether of ‘opening up’, ‘widening’, and ‘empowering’ opportunities for citizen engagement, or of ‘deepening’ democratic practice. ‘Political space’ is not only something taken up, assumed, or filled, but something that can be created, opened, and reshaped [36,37,38,39]. This creates a new interaction between people and institutions, eliminates traditional boundaries, and enables and empowers people to become engaged, and to voice and provide resistance. Developing a positive place attachment creates many health benefits (better psychological health, lower stress level, satisfying social relationships, greater satisfaction with the physical environment), and more active community participation, such as greater social and political involvement in their communities [26,40]. The application of the PST in RN Macedonia could help to start to unleash some of these benefits.

The qualitative analysis identified more facts about the influence and sense of control in the following directions: whether citizens’ voices are heard and their needs addressed; ideas and requirements are taken into account when making decisions; whether there are services in local government or groups in the local community that enable people to be involved in decision making. Rating the topic with negative score (mean = 2.6), citizens expressed their dissatisfaction of the local government, a feeling that their voices are heard in the pre-election period only. Those statements are mainly based on their experience with the municipality administration thus far. Similar feelings/perceptions and a similar score (mean score = 3) have been observed in the Scottish pilot study, where people thought that local authorities did not take into the consideration their needs and views sufficiently when decisions were made. Citizens were dissatisfied with the minimal feedback from the council, NHS, and other public agencies, and from the limited resources [25].

The positive comments engendered by a participative tool such as the place standard encourage the administration to make greater use of the practice of “open days” with citizens, as well as forums in the planning and preparation of certain projects. They also encourage citizens to attend more open sessions of the municipal council. Many of the respondents thought that implementation of an online survey such as a PST questionnaire is a good example of the local authorities’ willingness to listen and learn the voices of the citizens, although only a few thought that their expressed expectations and proposals are likely to be implemented in the programs. According to the citizens, “local authorities should not only have ideas, but also a courage and willingness to improve the quality of life in the place”. Compared to Scottish experience, in the city of Skopje, the work of local community groups and associations (e.g., housing associations) is absent or insufficient.

A similar score between the Macedonian and Scottish neighborhoods was also found (mean score = 3.3) in work and local economy, leaving the impression that both populations share many common problems: lack of employment opportunities, particularly for young people, and lack of local businesses that can provide jobs for them. In the Macedonian case, at municipality level, there are mainly employment opportunities in the service sector (restaurants, cafes). There is an obvious connection between problems of employment locally and unemployment and economy at the national level. Lack of subsidies for starting new businesses, as well as adequate training for this, are also missing.

Public space and paths infrastructure, which is safe for walking and cycling, provides not only a cost-effective form of transport but it also has health benefits, reducing the risk of circulatory diseases, diabetes, and certain cancers. According to the world health organization (WHO), the inclusion and improvement in quality of pedestrian and bicycle paths encourages people to walk and cycle, thereby promoting a healthy lifestyle. Replacing car use by active transport (walking or cycling) improves physical activity and social interaction, particularly in lower socioeconomic groups, and reduces health inequalities [5]. Although cycling lines and walking routes in Skopje are extending year by year, such routes do not extend beyond the central area of the city, and suburban municipalities do not have any such routes. Respondents regarded low levels of care and maintenance, and routes which are poorly marked and poorly lit with an absence of trees as a major problem. Most of the time, the routes are occupied by cars, and citizens feel that priority is invariably given to cars. The routes are normally inaccessible for wheelchair users and parents with young children in strollers due to the high pavements. In scoring the topic moving around with 3.0, the citizens expressed clear dissatisfaction with this aspect of the places in which they live.

### 4.3. Different Methods, Different Responses

An online survey is a cost-effective, less time-consuming way to reach a large number of people when compared to alternatives such as a public meeting or participation in a focus group. It is considered to offer more privacy, and an opportunity to speak out about the problems anonymously. Yet, the focus group approach, when properly moderated, undoubtedly gives the opportunity for more detailed discussions. In the RN Macedonia study, it was found that there were differences in the rating of the PST topics (Table 5) between two approaches. Participants tended to give a higher score for all topics (except work and local economy) in focus groups when compared to the online survey, where they were willing to criticize and were generally more moderate.

Participants were mainly motivated to participate in the focus group by the opportunity to contribute actively to improvement of the quality of life in the places where they live. All participants supported the implementation of such an activity. Even though the PST was implemented only at municipality level, citizens of Karpos expressed a high level of identity and belonging and attachment with the place (mean = 5.1). They expressed a feeling of connectedness to Karpos and were familiar with the history and heritage of the place. Participants rated the topic facilities and amenities highly and expressed a strong feeling of safety (mean = 4.9).

Both groups expressed their dissatisfaction over the care and maintenance of the public spaces and facilities, especially the greenery (mean = 2.9), and low level of environmental awareness of the citizens was highlighted as a problem and contributor to this situation. As expected, a high score was registered for the topics play and recreation, natural space, and social contact (mean = 4.4; mean = 3.4, and mean = 4.4), a finding consistent with Karpos having the largest municipally-owned green space(s) in Skopje (the city park and mountain Vodno). Negative feedback was in relation to poor environmental hygiene (littering, dog fouling), similar to the Scottish case study. Well-designed and accessible green spaces encourage people to play, relax, and meet people. It offers protection against heat stress and other impacts of climate change, reduces air pollution and noise levels, enhances biodiversity in the cities, and even allows urban food production [5]. Disconnection with nature impacts negatively on mental and physical health. There is substantial evidence on the benefits of maintaining contact with nature. Specifically, it decreases health problems related to chronic stress and attentional fatigue (increased irritability, difficulty concentrating, and being prone to errors on cognitive tasks) [5] often associated with living in dense cities, lower heart rate, and lower levels of negative emotions and anxiety [41,42].

Alongside care and maintenance, the lowest rating recorded was for traffic and parking (mean = 2.9). Most of the comments related to the congested traffic, lack of parking lots close to shopping malls, and fast driving over the boulevards at night. Overcrowding, inappropriate urban planning, over-construction in recent years, and the poor traffic culture of drivers were identified as key contributing factors. As a result, participants identified the problem of poor air quality and noise as priority public health issues. Compared to online survey results, the focus group participants expressed a higher level of influence and sense of control at a local level (mean = 3.5).

In general, the place standard was perceived by the citizens as a useful tool in influencing policy in the places where they live. Additionally, they considered the tool as a good link with the local or central authorities, especially if they receive feedback after the implementation of the tool.

### 4.4. Limitations and Lessons Learnt from Implementation of Place Standard

Proper implementation of the place standard tool, especially within different contexts (socio-economic, political, cultural, tradition, etc.) very much relies on good organization and preparation in the planning phase and particularly where human resources are limited. Good governance and top-level support are crucial, but so too is time. A four-month period for implementation, as in this case, is insufficient to prepare and implement such a comprehensive tool and, more importantly, to engage the wider public. Prior knowledge by the participants/respondents is needed, as well as training of the interviewers and moderators. Namely, there has not been a tradition in RN Macedonia of engaging citizens in the type of consultation which the place standard encourages, and certainly not of giving citizens meaningful input to decisions which shape their neighborhoods. Four months is arguably an insufficient time frame in which to overcome such deficits. Implementation and piloting the place standard tool during the summer months had a significant impact on the response rate. Regular meetings with local authorities and their team members, and engaging experts from the sociology and human behavior field in all phases is also important.

Translation and adaptation of the questionnaire to the Macedonian language was necessary and enabled a better understanding of the questions and some terms (in Macedonian, for example, there is not an appropriate word for ‘Place’). Continuous communication campaigns using different media, in parallel with implementation of the online survey, appeared particularly beneficial in the Macedonian case. The visit of an international expert and the opportunity to share experiences with a variety of domestic stakeholders proved beneficial. In particular, the knowledge and insights of local administrators and decision makers have been expanded and this has opened a window for future cooperation. Although they are all theoretically surmountable, there are barriers and impediments to delivering healthier, more equal, and more sustainable places (the triple win) through applying the place standard approach At the legislative level, there is an absence of strict regulation to oblige the municipalities to seek more active involvement of citizens and other stakeholders when discussing and development of the local policy. On the political level, the degree of readiness and willingness of some mayors to support and implement a tool like the place standard in their municipality/city may, of itself, be a barrier. Perhaps the most important barriers to implementing the place standard approach (and deriving its benefits) may be cultural, social, and, in some cases, educational. Citizens (participants) often display limited knowledge and awareness about what constitutes a healthy place. Moreover, in RN Macedonia, there is limited tradition of what we have described as inter-sectoral collaboration—in essence, cooperation among residents and other players to address common problems or shared issues. Such intersectoral collaboration lies at the heart of shaping the healthier, more equal and sustainable places through the place standard approach.

Another barrier is a lack of continuity in financing of the process, which should ideally come from the local municipality budget and/or the organizations with governance responsibility for the place undergoing review. During the next stages of PST implementation in the country, all these factors are required to be analyzed and elaborated in more depth.

## 5. Conclusions

Despite substantial contextual differences between Scotland and RN Macedonia, this pilot study established that the concept of a place standard and, importantly, the PST, were relevant and useful in the RN Macedonia.

More widely, introducing the idea of a place standard approach and the application of the PST (in pilot form) in RN Macedonia is a “good news story”. It has demonstrated the practicality of the tool as a means to deliver on key strategic priorities such, as the UN sustainable development goals [43] and, in a European city context, the WHO healthy cities Copenhagen consensus [44]. Stakeholders, especially citizens, fully embraced PST as a useful tool in influencing urban policies in their place.

The outcomes of the study reflected that citizens are more satisfied with issues like ‘identity and belonging’, followed by ‘facilities and amenities’ and ‘feeling safe’, although this was not the case for disabled groups and parents of infants. In contrast, citizens are mostly concerned with the issue of ‘traffic and parking’ as well as ‘care and maintenance’.

The application of the PST revealed that cultural and social inertia to engagement and intersectoral collaboration can be overcome. Participants (citizens and branches of the municipality) were willing to participate in a process which is both intersectoral and collaborative. They showed capacity and willingness to engage in an effort to understand and shape a response to a shared issue. As we have suggested, the transformation of a culture embedded over time within the duration of a short pilot exercise was ambitious in the extreme. Yet, the exercise revealed that residents had concerns and held views about elements of place and were willing to express them. In parallel with this, some other issues which reflect the specific context in the country (noise, climate change, drinking water, and sanitation problems) should be addressed in the future process of adaptation of the tool.

Applicability and proper implementation of the tool very much relies on: good governance and top-level management support; excellent preparation and organization; careful interviewers and focus group moderators; realistic timeframes and prior-knowledge of the respondents.

Municipality executive and political leadership regard the PST tool as a highly useful approach when creating the next annual policy and budget plan of the municipality

With specific reference to inter-sectoral collaboration, the process of applying the PST generated a high level of cooperation and enthusiasm among different stakeholders across society. This seems a very positive starting point from which to change behavior and for designing policies which can improve health and wellbeing, equity, and sustainability in residential areas. In the future, the INHERIT triple win can be fostered by engaging with hard-to-reach groups, including those most vulnerable and less politically engaged.

The study has confirmed that the use of PST increased the level of knowledge and confidence among the citizens and their desire to be actively involved in the decision making process. It is a free resource and a useful product for all professionals and communities interested in improving the quality of places through effective collaboration. Its availability should be promoted more widely through national and international networks to enable increased awareness and application. This will assist with the creation of healthy places. Implementation of the PST should be a continuous process to ensure organizations remain fully appraised of the on-going and very often changing physical, economical, and social needs and aspirations of communities.

## Figures and Tables

**Figure 1 ijerph-17-00194-f001:**
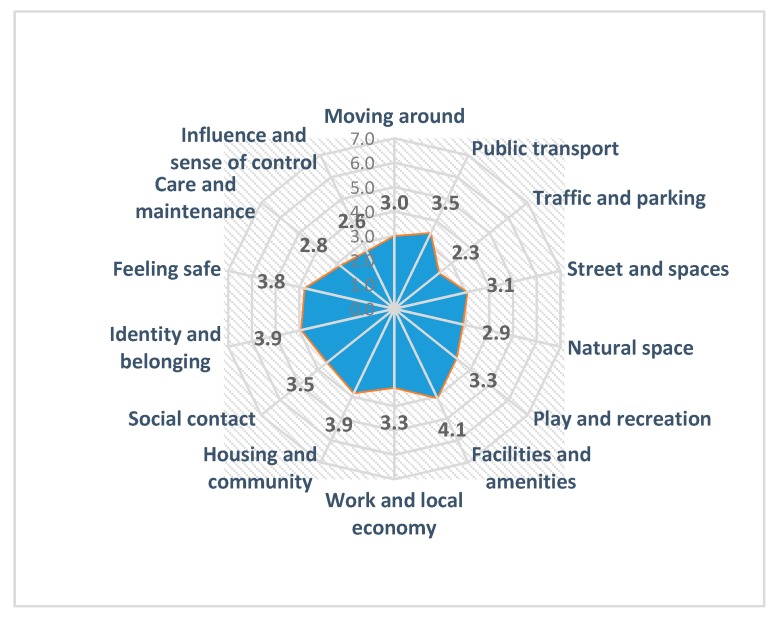
Summarized results of the online PST questionnaire from the pilot-study of the city of Skopje for the fourth-month period (2018–2019).

**Figure 2 ijerph-17-00194-f002:**
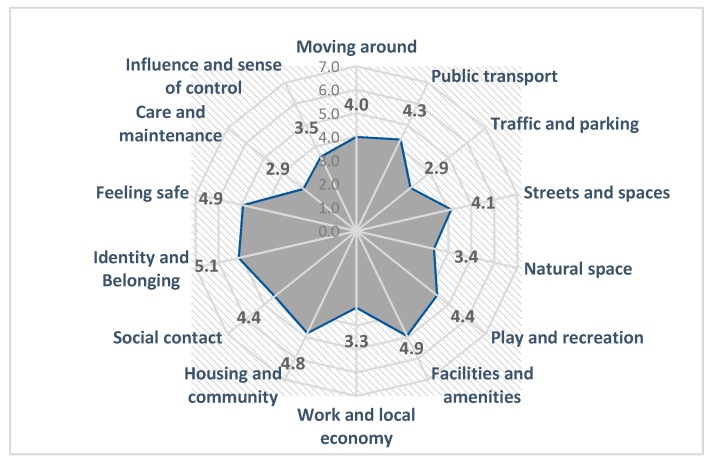
Summarized results of the citizen’s focus groups from the pilot-study of the city of Skopje, 2018.

**Table 1 ijerph-17-00194-t001:** Place standard tool topics—how good is our place?

Topic	Question
Moving around	Q_1_: Can I easily walk and cycle around using a good-quality routes?
Public Transport	Q_2_: Does public transport meet my needs?
Traffic and Parking	Q_3_: Do traffic and parking arrangements allow people to move around safely and meet community’s needs?
Street and Places	Q_4_: Do buildings, streets and public spaces create an attractive place that is easy to get around?
Natural Spaces	Q_5_: Can I regularly experience good-quality natural space?
Play and Recreation	Q_6_: Do I have access to a range of spaces and opportunities for play and recreation?
Facilities and Amenities	Q_7_: Do facilities and amenities meet my needs?
Work and Local Economy	Q_8_: Is there an active local economy and the opportunity to access good-quality work?
Housing and Community	Q_9_: Does housing support the needs of the community and contribute to a positive environment?
Social Interaction	Q_10_: Is there a range of spaces and opportunities to meet people?
Identity and Belonging	Q_11_: Does this place have a positive identity and do I feel I belong?
Feeling Safe	Q_12_: Do I feel safe?
Care and Maintenance	Q_13_: Are buildings and spaces well cared for?
Influence and Sense of Control	Q_14_: Do I feel able to participate in decisions and help change things for the better?

**Table 2 ijerph-17-00194-t002:** Basic demographic profile of respondents (*N* = 278).

Variable	Total *N* = 278 (Valid %)		
**Sex**			
Male	105 (38.8)		
Female	173 (62.2)		
**Ethnicity**		**Age group**	
Macedonians	247 (88.8)	Less than 16	1 (0.4)
Albanians	3 (1.1)	16–24	14 (5.0)
Turks	3 (1.1)	25–34	88 (31.7)
Roma	0 (0.0)	35–44	107 (38.5)
Vlachs	4 (1.4)	45–54	52 (18.7)
Serbs	6 (2.2)	55–64	16 (5.8)
Bosniaks	0 (0.0)	65 and over	0 (0.0)
Other	1 (0.4)		
Not declared	14 (5.0)		
**Municipality**	**total**	**m**	**f**
Aerodrom	36	18 (50.0)	18 (50.0)
Butel	5	2 (40.0)	3 (60.0)
Gazi Baba	20	6 (30.0)	14 (70.0)
Gjorce Petrov	10	6 (60.0)	4 (40.0)
Karpos	130	45 (34.6)	85 (65.4)
Kisela Voda	28	10 (35.7)	18 (64.3)
Saraj	0	0 (0.0)	0 (0.0)
Centar	38	13 (34.2)	25 (65.8)
Cair	10	4 (40.0)	6 (60.0)
Suto Orizari	0	0 (0.0)	0 (0.0)
Other	1	1 (100.0)	0 (0.0)

**Table 3 ijerph-17-00194-t003:** Testing the internal consistency of data (*N* = 278).

Variable	Total *N* = 278 (Valid 100%, Excluded 0.0%)						
Item	Mean	Min	Max	Range	Max/Min	Variance	N of Items
Item Means	3.280	2.349	4.079	1.730	1.737	0.269	14
Item Variances	2.779	2.141	3.337	1.196	1.559	0.165	14
Inter-Item Covariance’s	1.034	0.338	2.038	1.700	6.032	0.092	14
Inter-Item Correlations	0.371	0.150	0.671	0.521	4.482	0.008	14
**Cronbach’s Alpha**	0.892						14

**Table 4 ijerph-17-00194-t004:** Frequencies of scores for all fourteen topics of PST questionnaire.

Question	1		2		3		4		5		6		7		Average Score
	#	%	#	%	#	%	#	%	#	%	#	%	#	%	
Q1	63	22.7	52	18.7	56	20.1	54	19.4	42	15.1	8	2.9	3	1.1	2.99
Q2	41	14.7	51	18.3	49	17.6	58	20.9	45	16.2	25	9.0	9	3.2	3.45
Q3	106	38.1	70	25.2	46	16.5	26	9.4	18	6.5	9	3.2	3	1.1	2.35
Q4	69	24.8	58	20.9	50	18.0	31	11.2	33	11.9	26	9.4	11	4.0	3.08
Q5	73	26.3	65	23.4	44	15.8	42	15.1	31	11.2	15	5.4	8	2.9	2.89
Q6	55	19.8	45	16.2	58	20.9	51	18.3	38	13.7	22	7.9	9	3.2	3.27
Q7	30	10.8	36	12.9	40	14.4	45	16.2	57	20.5	43	15.5	27	9.7	4.08
Q8	48	17.3	43	15.5	61	21.9	67	24.1	39	14.0	16	5.8	4	1.4	3.25
Q9	30	10.8	39	14.0	46	16.5	55	19.8	57	20.5	37	13.3	14	5.0	3.85
Q10	51	18.3	35	12.6	45	16.2	63	22.7	42	15.1	30	10.8	12	4.3	3.53
Q11	29	10.4	41	14.7	44	15.8	53	19.1	51	18.3	39	14.0	21	7.6	3.92
Q12	45	16.2	32	11.5	39	14.0	49	17.6	58	20.9	45	16.2	10	3.6	3.78
Q13	63	22.7	67	24.1	51	18.3	58	20.9	27	9.7	11	4.0	1	0.4	2.84
Q14	89	32.0	65	23.4	51	18.3	35	12.6	22	7.9	6	2.2	10	3.6	2.62

**Table 5 ijerph-17-00194-t005:** Rating of the place standard topics implementing different approaches—online questionnaire/survey and focus groups.

Topic	Online Q/Survey	Focus Groups
Moving around	3.0	4.0
Public transport	3.5	4.3
Traffic and parking	2.3	2.9
Streets and spaces	3.1	4.1
Natural space	2.9	3.4
Play and recreation	3.3	4.4
Facilities and amenities	4.1	4.9
Work and local economy	3.3	3.3
Housing and community	3.9	4.8
Social contact and interaction	3.5	4.4
Identity and belonging	3.9	5.1
Feeling safe	3.8	4.9
Care and maintenance	2.8	2.9
Influence and sense of control	2.6	3.5

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
