# Peer review of "How Good Is our Place—Implementation of the Place Standard Tool in North Macedonia"

_ijerph, 2019, doi:10.3390/ijerph17010194_

Round 1
Reviewer 1 Report
The manuscript is well written. The results and arguments are well discussed and illustrated. There are no significant and crucial comments to it. However, for the betterment of the contents to be more understood by the audiences/readers, allow me to suggest that it would be good to add a graph/table on comparison on how the PST topics corelate with barriers mentioned/identified in part 4.4. (legislative level, political level, and social level) and proposed solution actions to that, so that the readers could grasp at the end, the whole picture of PST method from questions (status quo), barriers, and proposed solution actions; implemented in Macedonia. These all will give an additional value to the research results, and not just merely descriptive.
Author Response
Dear Sir/Madam,
in behalf of the team of authors, we want to express our gratefulness on all your comments, remarks and suggestions regarding the manuscript we have submitted to the Journal.
We find them very helpful indeed. And based upon them, we have tried to improve the text of our manuscript.
Our responses to your comments are as follows:
The manuscript is well written. The results and arguments are well discussed and illustrated. There are no significant and crucial comments to it. However, for the betterment of the contents to be more understood by the audiences/readers, allow me to suggest that it would be good to add a graph/table on comparison on how the PST topics corelate with barriers mentioned/identified in part 4.4. (legislative level, political level, and social level) and proposed solution actions to that, so that the readers could grasp at the end, the whole picture of PST method from questions (status quo), barriers, and proposed solution actions; implemented in Macedonia. These all will give an additional value to the research results, and not just merely descriptive.
Response: Small changes/clarifications have been made.
Reviewer 2 Report
Overall, this is a fair piece of work carried out on a relatively interesting topic. As described in detail below, however, the write up of the work does allow the reader to get a good grasp on what questions were examined as part of the research and what its key contributions are, seriously detracting from the overall quality of the work.
A description of the Place Standard should come much earlier in the text.
There is a very long introduction talking about the use of PST in Scotland before ever explaining what it actually is. The motivation for this review of the Scottish activity is unclear and very descriptive. I suggest re-writing it so that it provides motivation for the current work or cutting it altogether.
In part because of the above issues, the research questions and contributions of this work are not at all clear in the introduction.
The discussion of pro-environmental behaviour and moral norms (beginning on line 235) arises completely out of the blue. The reader was not aware that this was a topic that was being investigated in the current work as it has not up to this point been referred to. It is unclear what data this paragraph is based on. Unless rewritten to be much clearer I suggest that this paragraph be cut. Environmental behaviour comes in again on 276 but its links to the current work are not obvious.
In general, there is a need to make much clearer what this paper investigates and contributes. While the limitations and lessons learnt is an interesting short section, this is not written up as the main contribution of the paper. The cited aim (below) is not met by the current work.
'the aim of the Skopje case study was to understand how the tool can be implemented across different context - economic, social, cultural, and political'
Some examples of smaller issues:
The abstract could be written in a more concise and clear manner.
A session with the mayor is mentioned in the abstract but not at all in the text.
Line 26: unfinished sentence
Line 39: Quote missing reference with page number
Line 47: Again a quote without a reference
Line 48-49: unsupported
Line 80: the acronym PST already contains 'tool'
Line 82: 'poor....' arguably does not 'create' health and wellbeing
Line 136: characteristics- this error appears elsewhere
Line 181:There was a higher 181 proportion of Macedonian population (88.8%) in the study. Higher than what?
Table 4: not clear that this table is helpful- it is hard to read and presenting the distribution of the data graphically would be preferable.
Line 441: Causal language -The PST increased the level of knowledge and confidence among the citizens and their desire to 442 be actively involved in the decision-making process - features in the abstract and throughout the text. Given the research design, this is not appropriate.
Author Response
Dear Sir/Madam,
in behalf of the team of authors, we want to express our gratefulness on all your comments, remarks and suggestions regarding the manuscript we have submitted to the Journal.
We find them very helpful indeed. And based upon them, we have tried to improve the text of our manuscript.
Our responses to your comments are as follows:
Overall, this is a fair piece of work carried out on a relatively interesting topic. As described in detail below, however, the write up of the work does allow the reader to get a good grasp on what questions were examined as part of the research and what its key contributions are, seriously detracting from the overall quality of the work.
A description of the Place Standard should come much earlier in the text.
Response: Our preference would be to retain the current positioning. In order to explain this, we have added a sentence (line 229-232 in the revised version of the manuscript) saying “the paper describes the application of a relatively new tool for securing the involvement of communities in a comprehensive discussion of the merits and demerits of the places they live and particularly the influence this has on matters of health wellbeing and equity and through this, plotting the direction of change”, so we would argue against shifting the detailed description of the PST to the beginning. It would also disrupt the structure quite significantly
There is a very long introduction talking about the use of PST in Scotland before ever explaining what it actually is. The motivation for this review of the Scottish activity is unclear and very descriptive. I suggest re-writing it so that it provides motivation for the current work or cutting it altogether.
Response: We have adjusted accordingly.
In part because of the above issues, the research questions and contributions of this work are not at all clear in the introduction.
Response: Appropriate interventions and adjustments have been made in this regard.
The discussion of pro-environmental behavior and moral norms (beginning on line 235) arises completely out of the blue. The reader was not aware that this was a topic that was being investigated in the current work as it has not up to this point been referred to. It is unclear what data this paragraph is based on. Unless rewritten to be much clearer I suggest that this paragraph be cut. Environmental behavior comes in again on 276 but its links to the current work are not obvious.
Response: Some interventions have been made which we hope have kept the flow of the text.
In general, there is a need to make much clearer what this paper investigates and contributes. While the limitations and lessons learnt is an interesting short section, this is not written up as the main contribution of the paper. The cited aim (below) is not met by the current work.
'the aim of the Skopje case study was to understand how the tool can be implemented across different context - economic, social, cultural, and political'
Response: Appropriate adjustments have been made to outline aims more clearly and align them with the conclusions/discussions.
Some examples of smaller issues:
The abstract could be written in a more concise and clear manner. Resolved
A session with the mayor is mentioned in the abstract but not at all in the text. Resolved
Line 26: unfinished sentence Resolved
Line 39: Quote missing reference with page number. Resolved
Line 47: Again a quote without a reference
Response: “take-make-consume-dispose" is a phrase, not a quotation.
Line 48-49: unsupported
Response: This sentence and following one are supported by the same reference.
Line 80: the acronym PST already contains 'tool' Resolved
Line 82: 'poor....' arguably does not 'create' health and wellbeing. Resolved
Line 136: characteristics- this error appears elsewhere Resolved
Line 181: There was a higher 181 proportion of Macedonian population (88.8%) in the study. Higher than what?
Response: Amended with "There was a higher proportion of Macedonian population (88.8%) in the study, than the other ethnical groups living in the city".
Table 4: not clear that this table is helpful- it is hard to read and presenting the distribution of the data graphically would be preferable.
Response: We disagree as we believe It will make a significant disruption of the text. In this case, the graphical presentation will make less efficient data visualization.
Line 441: Causal language -The PST increased the level of knowledge and confidence among the citizens and their desire to 442 be actively involved in the decision-making process - features in the abstract and throughout the text. Given the research design, this is not appropriate.
Response: Resolved and properly clarified in the text
Reviewer 3 Report
The paper by Gjorgjev et al. “How good is our place - implementation of the place standard tool in North Macedonia” described the use of Place Standard Tool (14 questions of different topics) to collect public opinions on various aspects of the cities they live in. They collected the data by online questionnaires(278 responses) and focus group interviews of selected population (16). Cronbach’s alpha tests indicated high internal consistency of the 14 questions. The pool results showed participants were least satisfied with “traffic and parking” and most comfortable with “facilities and amenities” or “identity and belonging”. The authors also compared the results from the online survey and focus group. Both groups showed consistency in ranking the 14 topics, while the focus group typically gave higher scores. The paper is quite well written. I recommend publication after my concerns are addressed.
My biggest concern is on how the 14 topics are asked and the use of same scales (1 least satisfied to 7 most satisfied) for all of them. This setting implies equal weight on each topic and does not take into account the participant’s priorities in deciding how good the place is. Why not ask the participants to rank the 14 topics first and then rate each of them? This could put different weights on the 14 topics and also help study priority differences between each individual.
L26 “in a meaning”
Word missing.
L58 “… mortality.”
Please add a reference.
L174 “a data quantification protocol was used.”
Please add the details.
Author Response
Dear Sir/Madam,
in behalf of the team of authors, we want to express our gratefulness on all your comments, remarks and suggestions regarding the manuscript we have submitted to the Journal.
We find them very helpful indeed. And based upon them, we have tried to improve the text of our manuscript.
Our responses to your comments are as follows:
publication after my concerns are addressed.
My biggest concern is on how the 14 topics are asked and the use of same scales (1 least satisfied to 7 most satisfied) for all of them. This setting implies equal weight on each topic and does not take into account the participant’s priorities in deciding how good the place is. Why not ask the participants to rank the 14 topics first and then rate each of them? This could put different weights on the 14 topics and also help study priority differences between each individual.
Response: We would not agree with this viewpoint and trust the following explains this position.
(1) The aim of the study was to test the original tool and process. (2) The application of the tool in effect prioritizes the themes through a systematic process of considering all dimensions of a place. The PST breaks this down into 14. (3) The designers believe that only after this process can an individual or group be fully informed and determine what theme or themes are the greatest priority. (4) In the Skopje case study this process was followed and themes were prioritized. (5) The process can confirm or challenge the participants’ original perceptions of a place’s issues and assets. (6) A participant can start at any point of the wheel and they may start from what they consider to be the biggest concern. (7) However all areas must be considered given the co-dependency of each.
L26 “in a meaning”
Word missing. Resolved
L58 “… mortality.”
Please add a reference. Resolved
L174 “a data quantification protocol was used.”
Please add the details. Resolved
